# Multiple Asparaginase Infusions Cause Increasingly Severe Acute Hyperammonemia

**DOI:** 10.3390/medsci10030043

**Published:** 2022-08-12

**Authors:** Randal K Buddington, Karyl K Buddington, Scott C. Howard

**Affiliations:** 1Resonance, Arlington, TN 38002, USA; 2College of Health Studies, University of Memphis, Memphis, TN 38152, USA; 3Molecular and Cellular Physiology, LSU Health Sciences, Shreveport, LA 71104, USA; 4Department of Biological Sciences, University of Memphis, Memphis, TN 38152, USA; 5College of Nursing, University of Tennessee Health Science Center, Memphis, TN 38163, USA

**Keywords:** asparaginase, hyperammonemia, infusion reactions, chemotherapy, pig

## Abstract

Adverse reactions during and shortly after infusing asparaginase for the treatment of acute lymphoblastic leukemia can increase in severity with later doses, limiting further use and increasing relapse risk. Although asparaginase is associated with hyperammonemia, the magnitude of the increase in serum ammonia immediately after the infusion and in response to multiple infusions has not been examined. The concurrence of hyperammonemia and infusion reactions was studied using weaned juvenile pigs that received 12 infusions of Erwinia asparaginase (Erwinase; 1250 U/kg) over 28 days, with two 5-day recovery periods without asparaginase after the eighth and eleventh doses. Infusion reactions and prolonged hyperammonemia (>50 µM ammonia 48 h after the infusion) began after the fourth dose and increased with later doses. Dense sampling for 60 min revealed an acute phase of hyperammonemia that peaked within 20 min after starting the first infusion (298 + 62 µM) and lasted less than 1 h, without apparent symptoms. A pronounced acute hyperammonemia after the final infusion (1260 + 250 µM) coincided with severe symptoms and one mortality during the infusion. The previously unrecognized acute phase of hyperammonemia associated with asparaginase infusion coincides with infusion reactions. The juvenile pig is a translational animal model for understanding the causes of acute and chronic hyperammonemia, differentiating from hypersensitivity reactions, and for improving infusion protocols to reduce acute hyperammonemia and to allow the continued use of asparaginase.

## 1. Introduction

Asparaginase is a mainstay of chemotherapy for acute lymphoblastic leukemia (ALL) but is one of the top five therapeutic compounds known to cause adverse drug reactions [1]. The adverse infusion reactions associated with asparaginase mostly occur within the first hour, the incidence and severity of them increase with continued dosing, [2] and, when severe, can lead to omission of subsequent asparaginase doses and reduce event-free survival [3,4]. The forms of asparaginase commonly used include native *Escherichia coli* (*E. coli*) asparaginase, *E. coli* asparaginase conjugated with polyethylene glycol (pegaspargase), and *Erwinia* asparaginase (Erwinase), each of which have different antigenic properties, but all are known to cause infusion reactions. When infusion reactions limit further use of the native and pegylated forms of asparaginase derived from *E. coli*, [5,6,7] patients can be switched to Erwinase to continue therapy, but if they react to all available forms, then they end up receiving less asparaginase than indicated by their treatment protocol and have an increased risk of relapse [3]. 

The majority of asparaginase infusion reactions are attributed to hypersensitivity, although some patients without anti-asparaginase antibodies have “allergic-like reactions” [8] that resemble hypersensitivity [8,9]. Notably, hyperammonemia caused by the deamination of asparagine and glutamine and the ensuing catabolism of other amino acids [10] has been associated with asparaginase since the 1970s. The elevated ammonia is known to elicit symptoms that can be similar to hypersensitivity reactions [11,12]. The potential contribution of hyperammonemia to adverse reactions was not fully recognized until after 2010, when the intravenous administration of asparaginase became increasingly common [2,12,13]. 

An increase in blood ammonia during and immediately after the infusion has been used as a proxy for asparaginase activity [14]. However, the onset, magnitude, and duration of the rapid increase in blood ammonia, herein termed acute hyperammonemia, and the potential contribution to the adverse reactions that occur during or shortly after asparaginase infusion are uncertain. The study reported herein tracked the severity of infusion reactions and hyperammonemia resulting from providing 12 infusions of intravenous Erwinia asparaginase, using juvenile pigs used as a translational model for pediatric clinical patients. Understanding the contribution of hyperammonemia to infusion reactions is necessary to make decisions about continuing the first-line asparaginase or switching to an alternative asparaginase or infusion protocol to assure the successful completion of all needed asparaginase doses and to maximize event-free survival.

## 2. Materials and Methods

### 2.1. Pigs and Procedures

Newly weaned female pigs (~21 days of age, 6–8 kg; *n* = 6) were purchased from a commercial producer, transported to USDA-inspected and approved animal facilities, and housed in environmentally controlled conditions (22–24 C; 12:12 light:dark cycle). All phases of the research involving the pigs were approved by the University of Memphis Institutional Animal Care and Use Committee.

The pigs were transitioned over 2 weeks from a pig chow to a premium dog food (Purina One, St. Louis, MO, USA) with 28% protein, 13% fat, and 6% fiber with energy and nutrient levels that exceed the requirements of growing pigs. This was to mimic the increased nutritional support as performed when managing pediatric oncology patients. After the adaptation period, the pigs were fasted overnight and the following morning were sedated (Telazol, Parsippany, NJ, USA; 0.03 mg/kg, IM), general anesthesia was induced and maintained (Isoflurane 1–2%), and the pigs were prepared for aseptic surgery. All pigs had a catheter (3.5 Fr 25 cm umbilical vessel catheter; Umbilicath, Utah Medical Products, Inc., Midvale, UT, USA) placed in the jugular, exteriorized between the shoulders, and secured using a Stat-Lock (Bard Access Systems, Inc., Salt Lake City, UT, USA). The jugular catheter was used for the administration of Erwinia asparaginase (Erwinase, Jazz Pharmaceuticals, Palo Alto, CA, USA), other intravenous medicines, and for the collection of blood samples. Analgesia was provided for two days during the recovery (Meloxicam, Boehringer Ingelheim Animal Health USA Inc., Duluth, GA, USA; 0.75 mg, per os). The pigs received intravenous prophylactic antibiotic for 5 days post-surgery (Cefazolin, B. Braun Medical Inc., Irvine, CA, USA; 50 mg/kg). Pigs with infection at surgical sites were also treated with enrofloxacin (Baytril; Bayer AG, Whippany, NJ, USA; 2.5 mg/kg) or gentamycin (5 mg; generic), as directed by the collaborating veterinarian. Meloxicam was administered to pigs with signs of pain or discomfort.

Pigs with reduced appetite and lack of weight gain had Ensure (Abbott Laboratories, Abbott Park, IL, USA) added to the dry food to increase palatability and consumption. Supplemental parenteral nutrition solution was provided for two days to one pig to address low food consumption and to prevent weight loss. Transient dehydration was suspected for two pigs based on observations of reduced drinking and urine production. This was addressed by providing Gatorade (PepsiCo, Purchase, NY, USA) orally or via the duodenal catheter. Additional fluids (lactated Ringers, normal saline) were administered intravenously if warranted for dehydration. 

Administration of Erwinia asparaginase (Erwinase, Jazz Pharmaceuticals, Palo Alto, CA, USA; 1250 U/kg, equivalent to a human dose of 25,000 U/m^2^) was started 3–4 days after surgery via the jugular catheter. The pigs were provided food before and during the period of Erwinase administration. Observations of eating intensity and general activity were made before, during, and for 6 h after administration of Erwinase. The first 8 doses were provided every 48 h (Table 1). A subsequent recovery period of 5 days was then included to mimic the interruption in asparaginase administration that can occur with some protocols. Daily prednisolone (2 mg/kg; via the duodenal catheter) was started after the 7th infusion to suppress potential hypersensitivity reactions that were increasing in severity and was continued for the remainder of the study. The severe reactions elicited by the 9th dose, despite the administration of prednisolone, led us to provide epinephrine (0.05 mg IV at the time of dosing) for possible anaphylaxis [15,16].

The 10th dose (booster) was provided 24 h after the 9th dose to determine if the adverse reactions to the 9th dose would be replicated. The 11th dose was administered 48 h later, and the 12th dose was administered after another intervening period of 5 days.

Initially, the entire Erwinase dose was delivered evenly over a 10 min period. Due to the increasing severity of infusion reactions (initially lethargy, leading to vomiting, urticaria, loss of coordination, and eventually seizures), the protocol was revised after the 7th dose. Thereafter, 1.5% of the dose was administered over a 1-min period in combination with the epinephrine. After a 5-min delay, the remainder of the dose was infused during the following 10 min. 

Blood samples were collected in ethylenediaminetetraacetic acid tubes before each dose. For the 3rd through 7th doses, another sample was taken 60 min after infusing the Erwinase. For the first and 12th (final) doses, blood samples were collected 5, 10, 15, 20 30, and 60 min after completion of the Erwinase administration. Samples were immediately placed in ice and transported to the laboratory for processing and analysis. 

### 2.2. Analysis of Samples

Within 60 min after collection, the blood samples were centrifuged at 4 °C. Samples of plasma were used immediately to measure ammonia (Ammonia Colorimetric Assay Kit, BioVision; Milpitas, CA, USA), following supplier instructions.

Asparaginase activity was measured in plasma samples collected 48 h after infusing the 3rd, 5th, 6th, and 11th doses of Erwinase and for the series of plasma samples collected after infusion of the 12th dose. Activity was based on the rate of ammonia production after adding aliquots of plasma to 37 °C mammalian Ringers with 5 mmol asparagine. The rate of ammonia production was compared to a standard prepared with 0 to 50 U/mL of Erwinase. The standards were linear over this range (r = 0.99 or greater). Asparaginase activity was expressed as U/mL of plasma.

### 2.3. Statistics

Paired t-tests were used to detect differences in ammonia levels before and after infusing asparaginase. *p*-values less than 0.05 were considered significant.

## 3. Results

### 3.1. General Observations

Two pigs did not complete the study. The jugular catheter came out of one pig after the seventh infusion. Another pig was euthanized after the ninth dose of Erwinase due to progressive and intractable morbidity from bloating. The remaining four pigs received all 12 doses of Erwinase. However, one pig died 10 min after administering the initial 1.5% of the 12th and final dose; the remainder of the dose was not provided due to severe adverse reaction. The necropsy revealed hemorrhagic lungs and petechial hemorrhage in other organs.

### 3.2. Growth

Weight gain for the entire 28-day period averaged 119 g/day (±102). This included the minimal weight gain for 10 to 12 days after surgery (Figure 1). The rate of growth is lower than the nearly 400 g/d gained by healthy weaned pigs of similar age. 

### 3.3. Infusion Reactions

The first noticeable infusion reactions occurred after the fourth dose for three of the six pigs, with lethargy as the main observation (Table 1). The symptoms became progressively more severe with additional doses, with vomiting, anxiety, loss of balance, seizures, urticaria, respiratory distress, and the death of one pig before completing the 12th dose. Symptoms were transient, and the pigs were eating and behaving normally within 1–2 h after the dose. No symptoms were observed during the 5-day periods between the eighth and ninth doses and the eleventh and twelfth doses.

Despite protocol changes and the daily administration of prednisolone after the seventh infusion to reduce potential hypersensitivity, the ninth dose elicited severe symptoms. The additional precaution of intravenous epinephrine after the ninth dose did not prevent seizures in three of the four pigs that received all 12 planned doses and the death of one pig before completing the infusion. Only one pig did not have severe symptoms in response to the 12th dose.

### 3.4. Ammonia

Baseline ammonia levels for the six pigs before the initiation of Erwinase therapy averaged 16.7 µmol/L ± 8.1. Similar values were measured immediately before the infusion of the first three doses of Erwinase and for the intervening days (17.0 µM ± 8.7 at 24 h after the infusion). Plasma ammonia 60 min after the third infusion was 60 µM ± 26. The 60-min post infusion values increased to 279 µM ± 52 after the fourth infusion and averaged 599 µM ± 172 for the fifth through seventh infusions. Ammonia levels measured 48 h after infusing the fourth through eighth doses and immediately prior to the next dose continued to exceed 50 µmol/L (71 µM ± 30), indicative of persistent hyperammonemia. 

The initial dose of Erwinase resulted in a peak ammonia level that averaged 298 µM (±62) 20 min after starting the infusion and by 70 min was less than 100 µM (Figure 2). The 12th and final dose resulted in a peak that was significantly higher and remained higher compared with the first dose (*p*-values all < 0.01), corresponding with the increased severity of symptoms. 

### 3.5. Asparaginase Activity

Plasma collected prior to starting the 12th dose infusion protocol did not have asparaginase activity based on lack of ammonia production. Plasma samples collected 48 h after the third dose averaged 88 U/mL before infusing the fourth dose (Table 2). The low levels of asparaginase activity 48 h after the fourth, fifth, and tenth infusions are indicative of neutralization. Asparaginase activity was not detected 5 days after the 11th dose, which is not surprising given the short half-life of Erwinia asparaginase. Asparaginase levels declined rapidly after infusing the 12th dose (Figure 3).

## 4. Discussion

The increasing severity of infusion reactions of young pigs during and immediately after the administration of Erwinia asparaginase mimics the infusion reactions associated with hyperammonemia of patients treated with Erwinase [12] and both native and pegylated forms of *E. coli* asparaginase [2,13]. Notably, as we experienced during the infusion of the ninth and twelfth doses, the severities of patient responses are of greater magnitude later in therapy and particularly when reintroduced after a break in therapy [17]. The concordance of the severity of infusion reactions with the increasing magnitude of acute hyperammonemia during the first hour after starting the infusion is a novel finding that suggests hyperammonemia contributes to infusion reactions. Although hypersensitivity to Erwinase is less frequent than for the native and pegylated forms of *E. coli* asparaginase, the incidence of hyperammonemia may be higher [6,12], corresponding to a higher use of antiemetics [18]. The present findings emphasize that distinguishing between infusion reactions caused by hypersensitivity and acute hyperammonemia [19] is essential for decisions about interventions that will allow more patients to complete therapy and achieve remission.

### 4.1. The Acute and Prolonged Phases of Hyperammonemia

Our findings with pigs, reports from clinical studies, and dogs treated for lymphoma or leukemia [20] led us to propose a model for asparaginase-induced hyperammonemia that includes two phases (Figure 4). The dense sampling protocol we used for the first and last infusions revealed an acute phase of hyperammonemia characterized by ammonia concentrations that peaked within 20 min after starting the infusion then declined. This is consistent with a rapid production of ammonia from circulating asparagine and glutamine that overwhelms homeostatic capacities. The magnitude, timing, and transient nature of the acute phase has not been previously characterized and is consistent with the timing of onset, severity, and rapid recovery of the pigs from the infusion reactions. Notably, within 1 to 2 h after the infusion, ammonia levels had declined from the peak and the pigs had recovered, appeared normal, and were feeding. Importantly, waiting for just 2 h after the start of the infusion to measure ammonia levels will underestimate the peak ammonia concentration that occurs earlier. Decreasing the infusion rate or delaying the administration of the remaining dose has been proposed when infusion reactions coincide with hyperammonemia [19]. This allows symptoms to subside, typically within 20–30 min, and patients feel much better, even after resuming the suspended asparaginase infusion.

The prolonged phase of hyperammonemia is characterized by ammonia concentrations that exceed 50 µM for several days after the administration of asparaginase before declining in pigs (present study). This mimics the transient hyperammonemia in pediatric oncology patients treated with asparaginase for acute lymphoblastic leukemia [21] and non-Hodgkin lymphoma patients [21,22]. Ammonia production during the prolonged phase is determined by the activity of asparaginase and the endogenous synthesis and release of asparagine and glutamine into the blood. Prolonged hyperammonemia is of concern because of neuroinflammation and cognitive impairment [13,23], contributing to the CNS depression known as “asparaginase blues” [24], and the concern may be even greater for the developing brains of children. Moreover, the chronic exposure of pigs to elevated ammonia alters patterns of gene expression in various organs [25], indicating that prolonged hyperammonemia can have widespread affects. 

The acute and prolonged phases of hyperammonemia are partly explained by a combination of asparaginase activity, concentrations of substrates (asparagine and glutamine), and enzyme specificity. The affinity constants (Km) of Erwinase [26] for asparagine and glutamine are 48 µM and 360 µM, respectively, compared with the 95–100 µM and 520 µM concentrations of plasma asparagine and glutamine measured in pigs of similar age [27] and men [28]. Based on a blood volume of 7% of body weight, the infusion of 1250 U/kg of Erwinase and estimated 17,500 U/L of blood would rapidly deaminate circulating asparagine and glutamine. The half-life of Erwinase is shorter when infused [29], and a two-compartment pharmacokinetic model for a 70-kg individual [30] includes half-lives of 3.5 h during the distribution phase and 19.6 h for the elimination phase. After 48 h and more than eight half-lives, less than 100 mU/mL would remain in the blood after infusion. As in children, the smaller size of the pigs may have resulted in even lower activity [31,32]. Inactivation due to immune-associated neutralization [33,34] accelerates the loss of activity [34], resulting in higher Asn and Gln at the next infusion and greater magnitude of acute hyperammonemia. The greater than 90% decline in Erwinase activity within 60 min after concluding the 12th infusion (Table 2) is consistent with inactivation.

The combination of diminished Erwinase activity before the next infusion [35,36] and low glutaminase specific activity (13% of asparaginase activity [26]) would allow plasma glutamine to at least partially recover [35]. Even though asparagine may remain depleted for 48 h, plasma glutamine concentrations may increase enough that the next infusion causes another acute hyperammonemia event. This is consistent with the increased magnitude of acute hyperammonemia and the severity of infuse reactions associated with the ninth and twelfth infusions following the 5-day periods after the eighth and eleventh infusions, whereas providing the tenth infusion 24 h after the ninth dose reduced the magnitude of acute hyperammonemia and adverse infusion reactions. These findings emphasized that the activity of Erwinase and other asparaginases should remain high enough to deplete both asparagine and glutamine between infusions to minimize hyperammonemia and associated toxicity reactions. The asparaginase-induced changes in ammonia homeostasis confound the use of plasma ammonia concentrations to monitor asparaginase activity [14,31]. Instead, because asparaginase is absent in human sera [37], the ex vivo production of ammonia [38] may be more reliable for monitoring in vivo asparaginase activity [7]. 

The increasing severity of acute hyperammonemia and especially the significant difference between the first and last infusions indicate that multiple infusions of asparaginase progressively disrupt ammonia homeostasis. Individuals with urea cycle defects are characterized by hyperammonemia [39] that can be fatal if asparaginase is administered [40]. With ureagenesis restricted to the liver, the hepatotoxicity of asparaginase [41] may compromise urea synthesis. Although normal or elevated blood urea nitrogen among patients with hyperammonemia associated with asparaginase [2] imply a functional urea cycle, diminished capacities may be inadequate to rapidly clear the released ammonia, resulting in the increasing severity of acute hyperammonemia. The exceedingly high acute hyperammonemia after the 12th dose suggests that the five-day recovery was insufficient to regain normal ammonia homeostatic capacity. 

### 4.2. Asparaginase Hypersensitivity Overlaps with Acute Hyperammonemia

Adverse infusion reactions associated with hypersensitivity develop during 60 min infusions, generally last 1–2 h, [16] overlapping with the acute phase of hyperammonemia that causes similar symptoms in patients [42]. The incidence and severity of immune reactions to Erwinase [43] appear to be lower than that caused by other asparaginases [33,34]. However, hypersensitivity may have contributed to the infusion reactions. Although the administration of prednisolone did not prevent the increasing severity of infusion reactions among the pigs, corticosteroid therapy does not prevent an immune response to asparaginase among patients [6]. The administration of epinephrine with the ninth through twelfth infusions for possible anaphylaxis [15,16] did not reduce the severity of reactions. However, the reduced acute hyperammonemia and severity of infusion reactions following the 10th infusion implies that hyperammonemia contributed to infusion reactions. Differentiation between the contributions of acute hyperammonemia and hypersensitivity to infusion reactions is needed for decisions about appropriate interventions to continue asparaginase therapy and to maximize event-free survival. 

## 5. Conclusions

Effective interventions for hyperammonemia require understanding how asparaginase disrupts ammonia homeostasis. Of particular importance is why the severities of the acute and prolonged phases of hyperammonemia increase with continued asparaginase therapy. Juvenile pigs are a valuable model for investigating asparaginase-induced acute and chronic hyperammonemia and differentiating from hypersensitivity and silent inactivation to infusion reactions associated with Erwinase and other asparaginases. Our findings highlight that understanding how asparaginase disrupts ammonia homeostasis is tantamount to improving infusion protocols and interventions that will reduce the magnitude and severity of hyperammonemia and associated infusion reactions that are independent of immune responses.

## Figures and Tables

**Figure 1 medsci-10-00043-f001:**
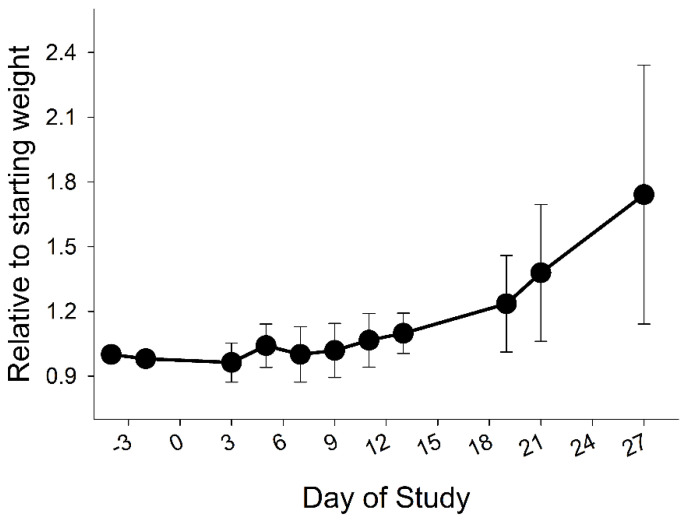
Relative changes in body weights of pigs. Body weights of the pigs were expressed relative to the initial weight before surgery (−4) and the start of the 28-day period with infusion of the 12 doses of Erwinase.

**Figure 2 medsci-10-00043-f002:**
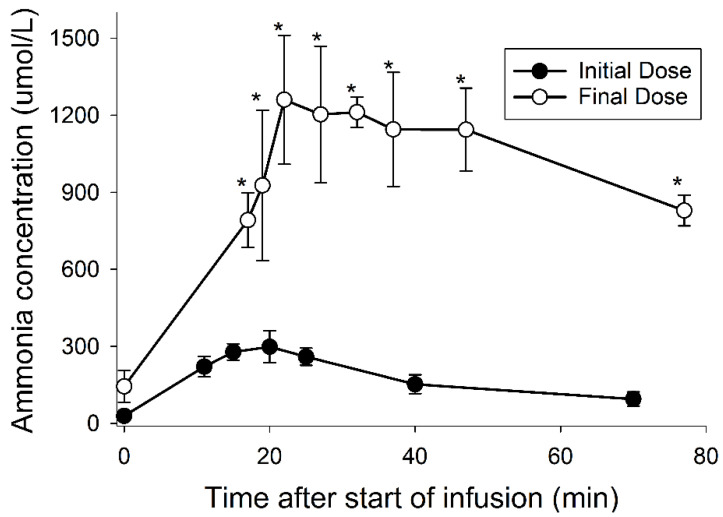
Acute hyperammonemia after the infusion of Erwinase. Plasma ammonia concentrations were measured before and at 1, 5, 10, 15, 20, 30, and 60 min after completing the infusion of the first and final doses of asparaginase. The offset for the final dose reflects the use of a longer (16 min) dosing protocol compared with the first infusion (10 min). The * indicate that significant differences were detected for comparisons between the first and final doses at the same times after completing the infusions.

**Figure 3 medsci-10-00043-f003:**
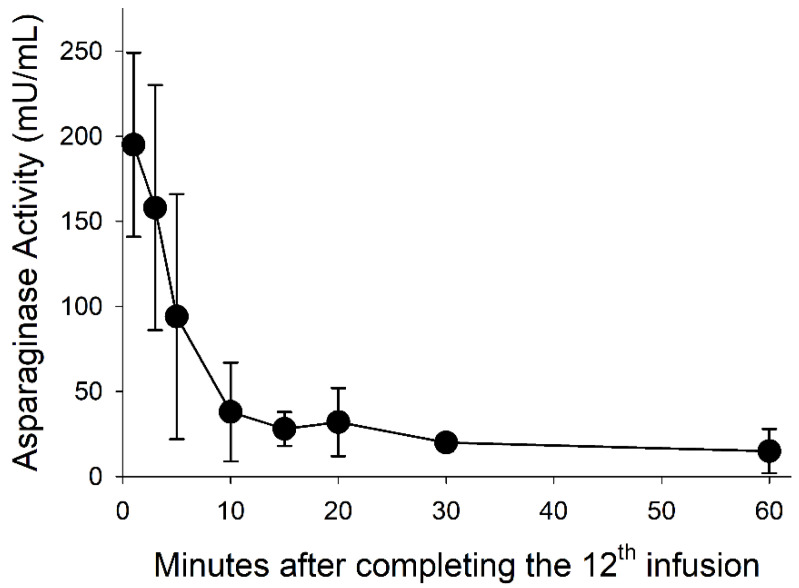
Post infusion asparaginase activity. Asparaginase activity declined during the 60 min following the 12th and final infusion.

**Figure 4 medsci-10-00043-f004:**
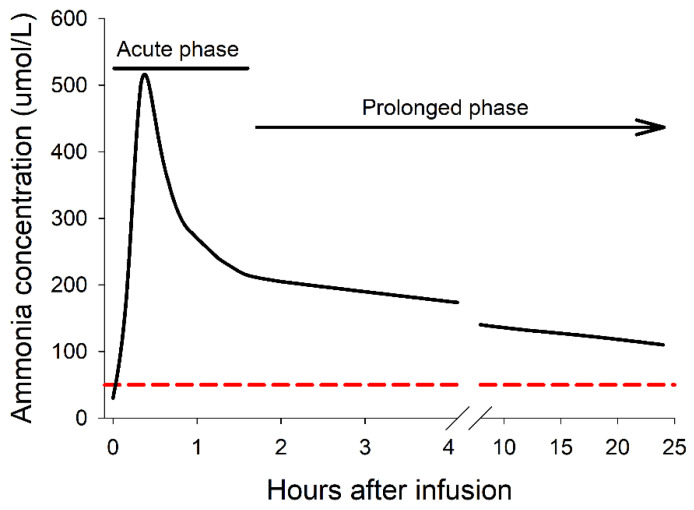
Model for acute and prolonged hyperammonemia associated with asparaginase infusion. Changes in ammonia concentrations after the infusion of Erwinase to the experimental pigs and those from clinical reports of patients treated with asparaginase were used to develop a model of hyperammonemia that included an early acute phase and a prolonged phase during which ammonia concentrations decline. The dashed red line indicates the upper limit of the normal range for ammonia concentration.

**Table 1 medsci-10-00043-t001:** Dosing schedule for the infusion of Erwinase with indications of the severity of symptoms and the use of prednisolone to reduce hypersensitivity and epinephrine for possible anaphylaxis.

Day of Study	Description	Symptom Severity *	Comments
−7, −5, −3	Pre-dose	0	Controls for baseline ammonia
1	Initial dose	0	
3	2nd dose	0	
5	3rd dose	1	Lethargy
7	4th dose	2	Lethargy, vomiting, discomfort
9	5th dose	2	Same
11	6th dose	3	
13	7th dose	3	Same; two with seizures, begin daily prednisolone administration (2 mg/m^2^ via duodenal catheter) for remainder of study to reduce hypersensitivity
15	8th dose	4	Severe symptoms; seizures, imbalance, urticaria
16–21	Recovery period	0	No symptoms; eating and gaining weight
22	9th dose	4	Severe symptoms Begin IV epinephrine (0.05 mg) at the time of each dose
23	10th dose (booster)	2–3	Mild to moderate symptoms
25	11th dose	3–4	Same severe symptoms with high respiratory rate, urticaria
26–29	Recovery period	0	No symptoms; eating and gaining weight
30	12th and final dose	4–5	Severe symptoms with one death

***** 0—no symptoms; 1—very mild change in behavior in response to the infusion; 2—obvious but considered moderate adverse reactions including urticaria, vomiting, signs of discomfort, difficulty walking; 3—mild seizures, vomiting, disorientation; 4—increased severity and number of symptoms, labored breathing; 5—symptoms considered to be life-threatening, requiring life-saving interventions (e.g., CPR, supplemental oxygen).

**Table 2 medsci-10-00043-t002:** Asparaginase activities (U/L) measured in plasma 48 h after infusing the third, fifth, sixth, and eleventh dose of 1250 U/kg.

Dose	Asparaginase Activity 48 h after Infusion (U/L)
3	88 ± 11
5	21 ± 12
6	10 ± 3
11	12 ± 3

## Data Availability

All of the data supporting the findings and conclusions of this study are presented in the results.

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
