# Peer review of "Multiple Asparaginase Infusions Cause Increasingly Severe Acute Hyperammonemia"

_medsci, 2022, doi:10.3390/medsci10030043_

Round 1

Reviewer 1 Report

This manuscript described using Juvenile pig models for investigating asparaginase induced acute and chronic hyperammonemia and differentiating from hypersensitivity, and silent inactivation to infusion reactions associated with Erwinase and other asparaginases. The findings suggesting asparaginase disrupts ammonia homeostasis is tantamount to improving infusion protocols and interventions that will reduce the magnitude and severity of hyperammonemia and associated infusion reactions that are independent of immune responses.

In conclusion, this manuscript contributes to understanding how asparaginase disrupts ammonia homeostasis which will be helpful to the researchers in the field. So, I am supportive of the manuscript for publication in Medical Sciences.

Author Response

We are grateful for the summary provided by the reviewer and appreciate their support for publication of our contribution in Medical Sciences.  The reviewer didn’t provide any suggestions for how we could improve our manuscript.  Changes in the manuscript are in response to comments provided by the second reviewer.

Reviewer 2 Report

In the present article, the authors have directed their efforts towards studying the event of hyperammonemia upon infusion of asparaginase, often used as a treatment for acute lymphoblastic leukemia. The authors interestingly identify two phases of hyperammonemia during their study- acute and prolonged- and how the infusion rate affects the ammonia homeostasis over a 30-day period spread across 12 doses. The understanding of the above may lead to incorporating modifications to the current infusion protocol in terms of the infusion rate and required intervention.

The study will have an elevated significance by comparing the Erwinase-treated and native or peg- E. coli asparaginase under the exact dosage and treatment conditions. Do different kinds of asparaginase treatment at different doses cause differences in the severity of the adverse reactions?

Have the authors considered looking at Glutamate and glutamine levels?

Was the dose maintained the same across the 12 doses?

Was there a correlation found between the varying levels of ammonia and the severity of the symptoms? A correlative analysis may be helpful.

 Does the ammonia levels drop below 50 umol/L during the prolonged phase with an increase in the period of time between the doses and if so, what other complications may account from an increased period between the doses lead to?

What role and the degree of contribution does each of the phases (acute and prolonged) of hyperammonemia have toward the hypersensitivity?

Have the authors considered looking into urea cycle abnormality-associated contribution to the severity of symptoms?

Page 9 lines 310-311: The incidence and severity…. Erwinase appears to be lower than that caused

Author Response

We thank the reviewer for their synopsis and comments.  One of us (SCH) has suggested to pediatric oncologists how the infusion protocol can be modified to reduce the acute hyperammonemia.  Doing so has reduced the incidence and severity of infusion reactions.  By publishing our findings in Medical Sciences more pediatric oncologists will be aware of the acute hyperammonemia and can modify infusion protocols accordingly.

The reviewer raises several important questions that require additional research to provide answers.  At the same time, these question highlight the importance of our discovery of acute and prolonged phases of hyperammonemia associated with infusion of asparaginase and the implications with infusion reactions. 

The study will have an elevated significance by comparing the Erwinase-treated and native or peg- E. coli asparaginase under the exact dosage and treatment conditions. Do different kinds of asparaginase treatment at different doses cause differences in the severity of the adverse reactions?

We agree but the scope and associated funding was restricted to Erwinase.  Our future studies will include comparisons of different forms of asparaginase, as well as IM vs infusion.  This initial study highlights the need for such studies. 

Have the authors considered looking at Glutamate and glutamine levels?  Future efforts will include a comprehensive evaluation of multiple amino acids, intermediates of the urea cycle, and evaluating immune responses to assess contribution of hypersensitivity to the infusion reactions and inactivation of the asparaginase.

Was the dose maintained the same across the 12 doses?  Yes but the protocol of administration was modified as described in attempts to reduce the severity of infusion reactions. 

Was there a correlation found between the varying levels of ammonia and the severity of the symptoms? A correlative analysis may be helpful.  We appreciate the comment of the reviewer.  We did not perform PK studies with each infusion.  As a result, we can’t determine if and how the increasing severity of infusion reactions among the pigs after specific infusions correlated with peak hyperammonemia.  The differences between the first and last administrations indicate there is such a relationship.  This needs to be examined in greater detail and will be in future studies.

 Does the ammonia levels drop below 50 umol/L during the prolonged phase with an increase in the period of time between the doses and if so, what other complications may account from an increased period between the doses lead to?  After the 5-day break, the ammonia levels prior to infusing the asparaginase were back below 50 umol/L.  However, with the later doses they remained above that level 48 hours after the previous infusion.  This is indicative of persistent hyperammonemia. 

The complications caused by prolonged exposure to elevated hyperammonemia have been reported for individuals with urea cycle defects.  Acute exposures are also characterized and match what we observed in the pigs.  What is not known are the responses to multiple exposures to hyperammonemia that occur with repeated doses of asparaginase.

What role and the degree of contribution does each of the phases (acute and prolonged) of hyperammonemia have toward the hypersensitivity?  Hyperammonemia doesn’t contribute to hypersensitivity.  Rather hyperammonemia causes symptoms that can be confused with hypersensitivity.  Differentiating between the contributions of hyperammonemia and hypersensitivity is important.  Understanding how to do so will require additional studies that include assessment of immune responses. 

Have the authors considered looking into urea cycle abnormality-associated contribution to the severity of symptoms?  This too will be the focus of future studies.    

Page 9 lines 310-311: The incidence and severity…. Erwinase appears to be lower than that caused…

Thank you for bringing this mistake to our attention.